# Spatiotemporal Variation and Influencing Factors of Grain Yield in Major Grain-Producing Counties: A Comparative Study of Two Provinces from China

**Zhipeng Wang** [1,2], **Ershen Zhang** [1] and **Guojun Chen** [1,*]

1 School of Urban Design, Wuhan University, Wuhan 430072, China; wangzhipeng@henu.edu.cn (Z.W.); eszhang@whu.edu.cn (E.Z.)
2 College of Geography and Environmental Science, Henan University, Kaifeng 475004, China
* Correspondence: guojunchen@whu.edu.cn

**Abstract:** The exploration of the spatiotemporal variations and influencing factors of grain yield in major grain-producing regions is greatly important to maintain stable and sustainable agriculture. Taking Henan Province and Heilongjiang Province as examples, this study reveals the spatiotemporal characteristics of grain yield at the county level by using multisource data on the economy, society, and natural geography from 2000 to 2021 and employing methods such as coefficients of variation, standard deviational ellipses, and spatial autocorrelation analysis. Moreover, geographical detector and geographically weighted regression models are combined to explore the differences in significant influencing factors between these provinces and the spatial heterogeneity of regression coefficients, respectively. The following findings are drawn: (1) Grain yield in both provinces gradually increased, with notable differences in the annual growth rate, the proportion, and at the county level. (2) The number of high-yield counties significantly increased and their spatial distribution became more concentrated, indicating a notable shift in the main regions. (3) The overall spatial correlation of grain yield steadily increased, and the local spatial correlation transitioned from random distribution to gradual aggregation. (4) There were significant differences in the influencing factors, where geographical environment, socio-economic factors, and input factors all affected both provinces. In summary, this study provides a scientific reference for governments worldwide to formulate rational and effective food production policies, thereby contributing to global food security and sustainable social development.

**Keywords:** grain yield; grain-producing region; spatiotemporal variation; influencing factor; county level; Heilongjiang Province; Henan Province; China





## 1. Introduction

Food security is the cornerstone of social development [1,2]. With only 7% of the global farmland, China feeds nearly one-fifth of the global population. Currently, China's self-sufficiency rate in food production has reached over 95%, making significant contributions to global food security [3,4]. However, with the rapid economic development and the acceleration of urbanization [5,6], there are growing tensions in the human–land relationship [7], increasing contradictions between agricultural development and the ecological environment [8], impediments to factor flows between urban and rural areas [9], and changes in livelihood capital for farmers [10], which pose significant threats to food production and food security. In addition, the outbreak of the COVID-19 pandemic hindered food transportation and affected food security [11–13]. Food security during the post-pandemic period still holds significant strategic significance for domestic economic recovery and social stability because China has a lower self-sufficiency of some staple crops, and the agricultural trade restrictions have a real impact on national food security.

In the face of issues such as imbalanced grain supply and demand with the restricted trade and transportation, China's government has ensured national food security by increasing grain yields [14–16], safeguarding food consumption [17], expanding grain storage [18–20], and enhancing grain trade [21,22]. As a basis for national stability [23,24], this yield is positively influenced by various underlying driving factors, such as farmland quality [25,26], agricultural mechanization input [27,28], fertilizer and pesticide use [29,30], agricultural subsidies [31,32], and urbanization [33,34]. However, abnormal weather conditions such as droughts, floods, and hailstorms can bring many negative effects [35–37]. Furthermore, with the gradual application of modern and digital technologies in agricultural production [38–42], smart agriculture plays an increasingly significant role in food security. The previous studies of influencing factors focused on the natural environment, socio-economic factors, or input factors, thus lacking a comprehensive consideration of different dimensions and hindering the quantitative identification of the direction and intensity of factors. Therefore, it is crucial to study the comprehensive mechanisms of multiple factors.

In terms of the research scale, previous studies mainly focused on large-scale areas such as nations [43,44], regions [45,46], or provinces [47,48], lacking a more precise level. As the most basic and complete administrative unit in China, the county-level grain yield can reflect the level of regional development more widely and fundamentally. However, the existing county-level studies have only concentrated on single, small regions, without discussing the interregional differences. Considering the significant variations in natural and social conditions across different regions, it is valuable to scientifically select county-level research areas and scales. As the two provinces with the highest grain yields in China, Henan Province and Heilongjiang Province exhibit significant differences in terms of social and economic development, agricultural practices, and geographical environments [49–52]. Thus, this study takes Henan Province and Heilongjiang Province as the research areas. Based on multisource data from 2000 to 2021, a comparative study of spatiotemporal evolution was conducted by employing coefficients of variation, standard deviational ellipses, and spatial autocorrelation. Moreover, this study utilized geographic detectors and geographic weighted regression to examine influencing factors, thereby contributing to provide recommendations for food security.

The organization of the remainder of this paper is as follows: Section 2 describes the data, the methods, and the indicator system. Section 3.1 presents the spatiotemporal characteristics of grain yield, while Section 3.2 analyzes the spatial correlation of grain yield. Section 3.3 utilizes geographic detectors and geographic weighted regression models to explore the similarities, differences, and spatial heterogeneity of the factors influencing grain yield. The discussions and recommendations are presented in Section 4. The conclusions are drawn in Section 5.

## 2. Materials and Methods

### 2.1. Description of the Study Areas

Henan Province, located in the middle part of China, lies between 31°23′ and 36°22′ N and between 110°21′ and 116°39′ E, with a land area of 165,700 square kilometers. The topography of Henan Province gradually slopes from west to east, with mountainous and hilly areas in the west and the Huanghuai Plain in the central and eastern parts. It has a transitional continental monsoon climate from the northern subtropical zone to the warm temperate zone, characterized by four distinct seasons and simultaneous rainfall and heat. The climate is relatively well-suited to crop growth, and there are two harvests per year. In 2021, the grain yield in Henan Province reached 65.42 million tons, accounting for 9.58% of the national total, ranking second among all provinces in China.

As the northernmost province with the highest latitude, Heilongjiang Province lies between 43°26′ and 53°33′ N and between 121°11′ and 135°05′ E, with a land area of 473,000 square kilometers. The topography of Heilongjiang Province with extensive plains is generally high in the northwest and southeast and low in the northeast and southwest.

With a cold temperate and temperate continental monsoon climate, the land is suitable for crop growth in summer but unsuitable in winter. Thus, there is only one agricultural harvest per year. In 2021, the grain yield in Heilongjiang reached 78.677 million tons, accounting for the highest proportion, at 11.52%, of the national yield.

In this study, the county-level administrative units of the above provinces were defined as research units. Since urban counties are mainly engaged in non-agricultural production, the administrative divisions of the urban counties were combined into one research unit in the analysis process. Based on the administrative divisions in 2021, a total of 120 sample areas in Henan and 75 research sample areas in Heilongjiang were obtained. Figure 1 shows the regional overview of Henan and Heilongjiang Provinces.

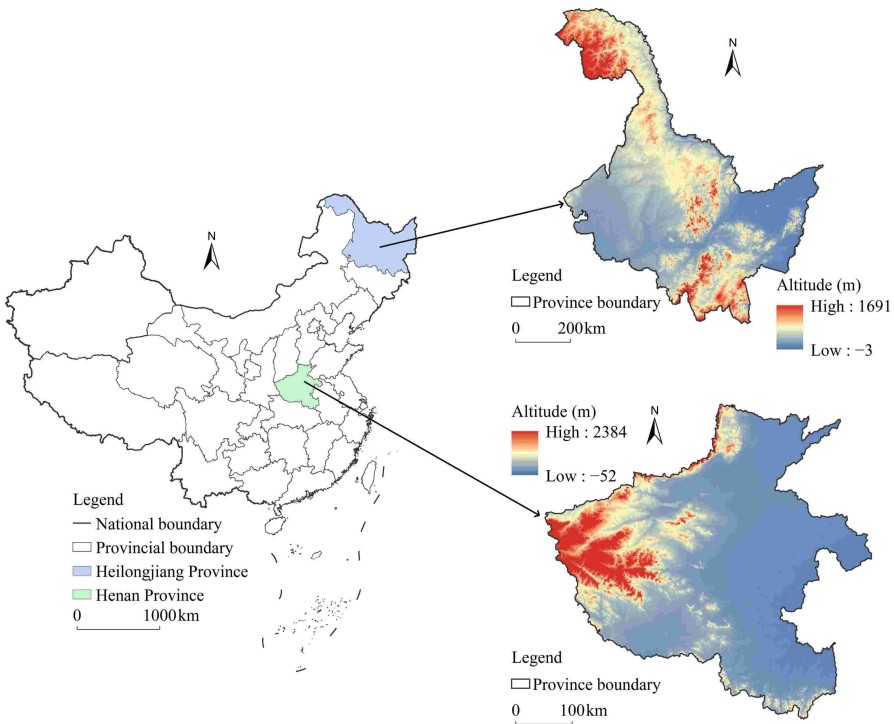

**Figure 1.** Overview map of Henan Province and Heilongjiang Province.

*2.2. Research Data*

2.2.1. Database

The data used in this study were as follows:

(1) Vector data included administrative region vector data from 120 research units in Henan and 75 research units in Heilongjiang, which were obtained from http://www.ngcc.cn/ngcc/ (accessed on 5 March 2023).

(2) Statistical data mainly included county-level grain yield, GDP, farmers' per capita net income, grain crop area, total power of agricultural machinery, net amount of fertilizer application, and effective irrigation area, derived from the statistical yearbook for the corresponding year, which were obtained from https://tjj.henan.gov.cn/ (accessed on 12 April 2022) and http://tjj.hlj.gov.cn/ (accessed on 18 June 2023).

(3) Meteorological observation data included precipitation, temperature, and sunshine hours at various meteorological stations in the two provinces, which were obtained from https://data.cma.cn (accessed on 2 February 2023).

(4) Digital elevation data (DEM) mainly included the digital elevation of the two provinces, derived from http://www.resdc.cn (accessed on 25 May 2023).

2.2.2. Data Processing

(1) Vector data processing. Based on the administrative division adjustment in 2021, the vector maps of the two provinces and counties were processed by using ArcGIS. The urban districts were merged into one urban district unit. The county and county-level cities remained unchanged, and the administrative region vector data were obtained.

(2) Statistical data processing. Since the urban district is an administrative unit, the data of each urban district is the sum of the statistical data of the urban districts. The data of each county and county-level city were published in the statistical yearbook.

(3) Meteorological data processing. Using the regional analysis tool of ArcGIS software, the data of each meteorological station were interpolated and smoothed, and partition statistics were generated based on the vector range of the county administrative region to obtain the average annual temperature, annual precipitation, and annual sunshine hours of each county in different years.

(4) Processing of digital elevation data. Using the slope analysis tool in ArcGIS software, the digital elevation data were first converted into slope data. Similarly, the average slope of each county was obtained by dividing the statistical data according to the vector range of the county administrative region.

*2.3. Indicator System*

In this study, county-level grain yield was considered as the dependent variable and incorporates factors related to the geographical environment, socio-economic conditions, and input factors were included as independent variables [53–57]. Specifically, the geographical environment factors include average annual temperature, annual precipitation, annual sunshine duration, and average slope [58–61]. The socio-economic factors included the proportion of agricultural labor force in the total population, GDP, the proportion of primary industry output value to GDP, and farmers' per capita net income [2,62–64]. The input factors included sown area of grain crops, the total agricultural machinery power, the pure amount of fertilizer application, and the effective irrigation area [65–68]. Descriptive statistics for these variables are presented in Table 1.

**Table 1.** Selection of influential factors indicators.

| Type | Indicator | Indicator Description | Unit |
|---|---|---|---|
| Geographical Environment Factors | Average Annual Temperature ($x_1$) | Average temperature recorded annually | Celsius (°C) |
| | Annual Precipitation ($x_2$) | Total precipitation received annually | Millimeters (mm) |
| | Annual Sunshine Duration ($x_3$) | Total hours of sunshine received annually | Hours |
| | Average Slope ($x_4$) | Average slope of the terrain in the study area | Degree (%) |
| Socio-economic Factors | Proportion of Agricultural Labor ($x_5$) | Proportion of the agricultural labor force to the total population | Percentage (%) |
| | GDP ($x_6$) | Gross domestic product of the region | Currency (e.g., USD, CNY) |
| | Proportion of Primary Industry GDP ($x_7$) | Proportion of the output value of the primary industry to GDP | Percentage (%) |
| | Per Capita Net Income of Farmers ($x_8$) | Average income earned by farmers per capita | Currency (e.g., USD, CNY) |

**Table 1.** *Cont.*

| Type | Indicator | Indicator Description | Unit |
|---|---|---|---|
| Input Factors | Sown Area of Grain Crops ($x_9$) | Total area dedicated to the cultivation of grain crops | Hectares (ha) |
| | Total Agricultural Machinery Power ($x_{10}$) | Combined power of all agricultural machinery in use | Kilowatts (kW) |
| | Pure Amount of Fertilizer Applied ($x_{11}$) | Total amount of fertilizer applied | Metric Tons (MT) |
| | Effective Irrigation Area ($x_{12}$) | Area of land effectively irrigated for crop cultivation | Hectares (ha) |

Note: The units provided in the table represent the standard units commonly used for the respective indicators.

*2.4. Research Framework*

Firstly, this study constructed an evaluation index system from three aspects: geographical environment, socio-economic factors, and factor inputs. Then, the county-level spatiotemporal changes were measured based on the coefficient of variation and standard deviation ellipse. Then, ESDA was applied to examine the county-level spatial agglomeration. Finally, geographic detectors and geographic weighted regression were used to analyze the regression relationship and spatial heterogeneity between grain yield and driving factors. The research framework is shown in Figure 2.

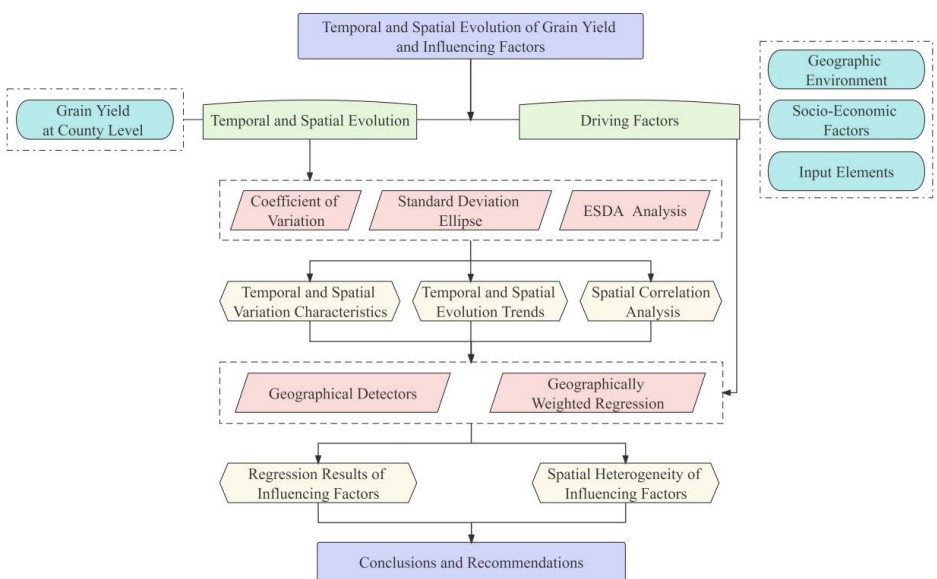

**Figure 2.** The research framework.

*2.5. Research Methods*

2.5.1. Coefficient of Variation

The coefficient of variation can objectively measure the degree of variation within a dataset. Compared to some measures such as range, variance, and standard deviation, the coefficient has the advantage of more accurately reflecting the dispersion of data [69,70]. At the county level, because of the numerous units, this indicator can avoid the bias caused by absolute differences and reflect the uneven distribution between different counties. The formula to calculate the coefficient of variation is as follows:

$$CV = \frac{1}{\overline{x}}\sqrt{\frac{1}{n-1}\sum_{i=1}^{n}(x_i - \overline{x})^2} \tag{1}$$

In the Equation (1), *CV* represents the coefficient of variation, *n* denotes the number of study units, $x_i$ represents the grain yield of the *i*th county, and $\overline{x}$ signifies the average yield across all counties.

### 2.5.2. Standard Deviation Ellipse

The standard deviation ellipse can provide a comprehensive depiction of the overall characteristics of the geographic distribution [71,72]. Parameters include ellipse area, centroid coordinates, major axis length, minor axis length, and orientation angle. The ellipse area represents the main distribution area, the centroid coordinates indicate the relative spatial position of geographic elements, the major axis represents the degree of concentrated influence of geographic elements in the main trend direction, the minor axis represents the degree of concentrated influence in the secondary direction, and the orientation angle represents the main trend direction of development. This study utilized the standard deviation ellipse to analyze the layout characteristics and evolution trends, with the following formula:

$$M(\overline{X}, \overline{Y}) = \left| \frac{\sum\limits_{i=1}^{n} w_i x_i}{\sum\limits_{i=1}^{n} w_i}, \frac{\sum\limits_{i=1}^{n} w_i y_i}{\sum\limits_{i=1}^{n} w_i} \right| \tag{2}$$

$$\tan\theta = \frac{\left[ \left( \sum\limits_{i=1}^{n} w_i^2 x_i'^2 - \sum\limits_{i=1}^{n} w_i^2 y_i'^2 \right) + \sqrt{ \left( \sum\limits_{i=1}^{n} w_i^2 x_i'^2 - \sum\limits_{i=1}^{n} w_i^2 y_i'^2 \right)^2 + 4 \left( \sum\limits_{i=1}^{n} w_i^2 x_i'^2 y_i'^2 \right) } \right]}{2 \sum\limits_{i=1}^{n} w_i^2 x_i' y_i'}. \tag{3}$$

$$\sigma_x = \frac{\sqrt{\sum\limits_{i=1}^{n} (w_i x_i' \cos\theta - w_i y_i' \sin\theta)^2}}{\sum\limits_{i=1}^{n} w_i^2}. \tag{4}$$

$$\sigma_y = \frac{\sqrt{\sum\limits_{i=1}^{n} (w_i x_i' \sin\theta - w_i y_i' \cos\theta)^2}}{\sum\limits_{i=1}^{n} w_i^2}. \tag{5}$$

In the formula, $M(\overline{X}, \overline{Y})$ represents the centroid coordinates, $\theta$ represents the rotation angle, which indicates the angle between the positive north direction and the clockwise rotation of the major axis, $x_i$ and $y_i$ are the coordinates of point features, $w_i$ represents the weights, $x_i'$ and $y_i'$ represent the coordinate deviations of point features from the centroid, and $\sigma_x$ and $\sigma_y$ represent the standard deviations along the *X* and *Y* axes.

### 2.5.3. Exploratory Spatial Data Analysis (ESDA) Analysis

Exploratory spatial data analysis (ESDA) as a visual and analytical approach can reveal the degree of similarity in many attribute values between the neighboring spatial units. For example, ESDA was used to investigate the spatial correlation characteristics in Henan and Heilongjiang [73,74]. The popular measures from ESDA are the Global Moran's Index and the Local Indicators of Spatial Association (LISA) cluster map.

$$I = \frac{\sum\limits_{i=1}^{n} \sum\limits_{j=1}^{n} \omega_{ij}(x_i - \overline{x})(x_j - \overline{x})}{S^2 \sum\limits_{i=1}^{n} \sum\limits_{j=1}^{n} \omega_{ij}} \tag{6}$$

$$S^2 = \frac{1}{n}\sum_{i=1}^{n}(x_i - \bar{x})^2. \tag{7}$$

In the formula, $I$ represents Moran's Index, $\omega_{ij}$ denotes the spatial weight between feature $i$ and feature $j$, and $x_i$ and $x_j$ are the observed values of feature $i$ and feature $j$. The range of $I$ is $[-1, 1]$, where a value less than 0 indicates a negative spatial correlation, a value equal to 0 indicates no spatial correlation, and a value greater than 0 indicates a positive spatial correlation.

A LISA cluster map can reveal the correlation of some attribute values between each observation location and its neighbors within a given study area. It is intended to show spatial clustering patterns of high and low values from a local perspective. Therefore, GeoDa software was used to create LISA cluster maps to analyze the local spatial autocorrelation and clustering phenomena in the counties of Henan and Heilongjiang Provinces.

### 2.5.4. Geographical Detector

The geographical detector model is an important tool for investigating spatial heterogeneity [75,76], including differentiation and factor detection, interaction detection, risk zone detection, and ecological detection. Among them, the differentiation and factor detection can analyze the explanatory degrees of different influencing factors on the spatial heterogeneity. The model is defined as follows:

$$q = 1 - \frac{\sum\limits_{h=1}^{L} N_h \sigma_h^2}{N\sigma^2} = 1 - \frac{SSW}{SST} \tag{8}$$

$$SSW = \sum_{h=1}^{L} N_h \sigma_h^2, SST = N\sigma^2. \tag{9}$$

where $q$ represents the explanatory power index; $h = 1, \ldots, L$ denotes the stratum or partition of variable $Y$ or factor $X$, which refers to the classification or zoning; $N_h$ and $N$ are the number of units in stratum $h$ and the entire region, respectively; $\sigma^2_h$ and $\sigma^2$ are the variance of $Y$ values in stratum $h$ and the entire region; $SSW$ and $SST$ represent the sum of squares of within-group variance and the total sum of squares, respectively; and the range of $q$ is $[0, 1]$, where $q$ closer to 1 for $x_i$ indicates a stronger explanatory power for the spatial heterogeneity, while a value closer to 0 indicates a weaker explanatory power. By comparing $q$ of different factors, we can detect the magnitude of their effects on the spatial heterogeneity.

### 2.5.5. Geographically Weighted Regression (GWR)

In order to further analyze the county-level influencing factors, we examined the causal relationships from both a conventional and spatially correlated perspective by employing a correlation analysis and GWR [77,78]. Correlation analysis reveals the degree of association between geographic factors and identifies the main factors. GWR can reveal the quantitative relationship between two or more factors and consider local characteristics as weights in the spatial suppression process, which is effective to explain the spatial variations in influencing factors. The model is as follows:

$$Y_i = \alpha_0(S_i, T_i) + \sum_{j=1}^{n} \alpha_j(S_i, T_i)x_{ij} + \varepsilon_i \tag{10}$$

In the equation, $(S_i, T_i)$ represents the spatial geographic coordinates of the $i$-th county unit (serving as the geographical weights), and $\alpha_j(S_i, T_i)$ represents the continuous function of the coefficient $\alpha_j(S_i, T_i)$ of the independent variable $x_k$ at the $i$-th county unit.

## 3. Results

### 3.1. Temporal and Spatial Variation Characteristics of Grain Yield

3.1.1. Temporal Variation Characteristics

The grain yield steadily increased and the production capacity greatly changed (Figure 3). There was an overall increase from 2000 to 2021, with a provincial-level variation. The yield in Henan increased from 41.015 million tons to 65.442 million tons, while in Heilongjiang, it surged from 25.455 million tons to 78.677 million tons with a significantly higher increment. From 2000 to 2009, Henan had more yield than Heilongjiang, but the difference gradually narrowed. Heilongjiang surpassed Henan in 2010 and then the gap increased. Henan Province experienced yield reductions in 2003 and 2021. Natural disasters such as major flooding in Huai River Basin and some autumn floods of the Yellow River were the main reasons, which had a detrimental impact on agricultural yield [79,80]. Heilongjiang also suffered yield reductions in 2003 and 2016 due to a severe drought in Northeast China and other extreme weather events [81].

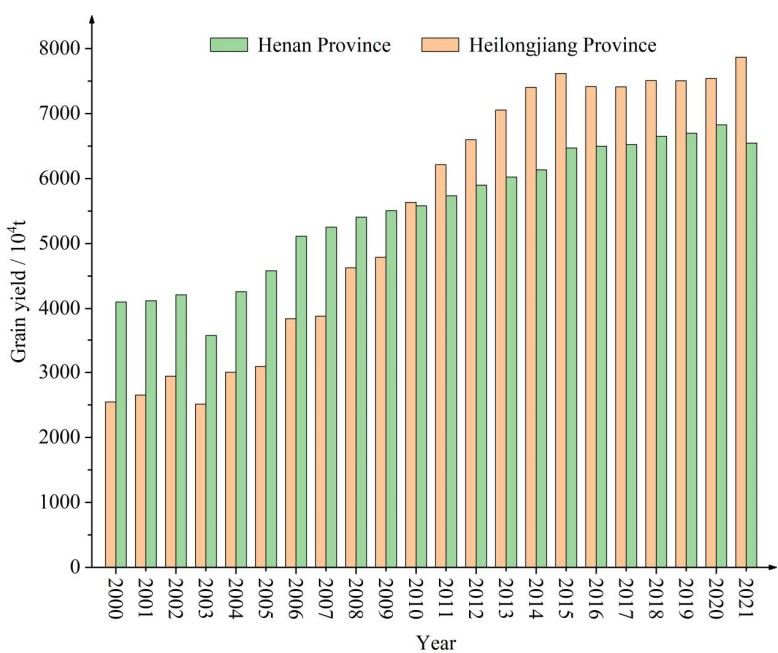

**Figure 3.** Changes in grain yield in Henan Province and Heilongjiang Province from 2000 to 2021.

The annual growth rate of grain yield fluctuated significantly, with notable differences between two provinces (Figure 4). The growth rates and average annual growth rates in Henan and Heilongjiang exhibited significant differences from 2000 to 2021, with distinct inter-annual and inter-decadal variations. The average annual growth rate in Henan was 2.36% while it was 5.80% in Heilongjiang, both higher than the national average annual growth rate of 1.97%. In terms of the temporal change, from 2000 to 2010, the annual growth rates in Henan and Heilongjiang displayed a large fluctuation, alternating between a substantial increment and decrement. From 2010 to 2021, the fluctuation magnitude in the annual growth rates in both provinces gradually reduced, with Heilongjiang exhibiting the greater variability. In terms of the inter-annual variations, Henan had the lowest (−15.21%) annual growth rate in 2003, while the highest was (19.35%) in 2004. This can be attributed to a significant decrement in 2003 [74]. Additionally, the severe floodings in 2021 led to a 4.13% decrease [80]. The lowest annual growth rate in Heilongjiang was −14.58% in 2003, while the highest was 24.30% in 2006 [81].

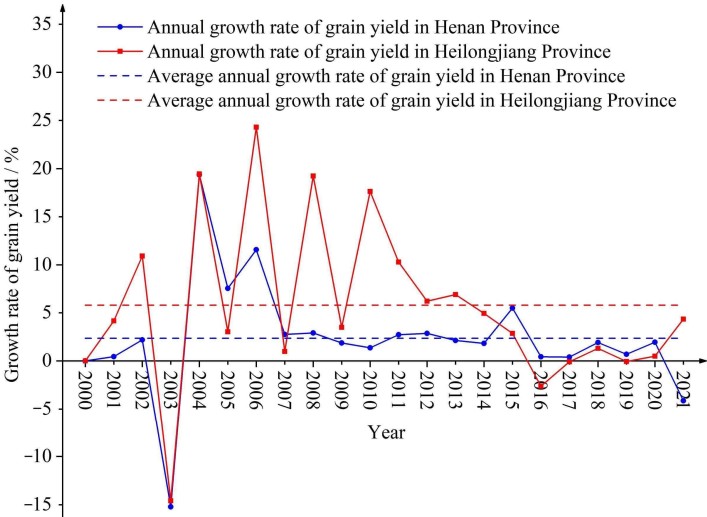

**Figure 4.** Changes in the growth rates of grain yield in Henan Province and Heilongjiang Province from 2000 to 2021.

The proportion of grain yield changed significantly (Figure 5), with a widening gap in contribution capacity. From 2000 to 2021, there were the significant proportion changes with different trends in both provinces. The proportion changes in Henan had two stages: a fluctuating increase stage (2000–2007) and a gradual decline stage (2007–2021). In the first stage, the total proportion in Henan had an approximate increasing trend, with the national proportion increasing from 8.87% to a peak of 10.42%. Then, the proportion fluctuated, but this was followed by an overall decreasing trend with a smaller change. Heilongjiang also had two stages: a sharp increase stage (2000–2014) with the national proportion increasing dramatically from 5.51% to 11.57%, and a fluctuating stage (2014–2021) with the fluctuating proportion consistently above 11% in a "W" pattern. In addition, the national proportion of Heilongjiang was steadily increasing.

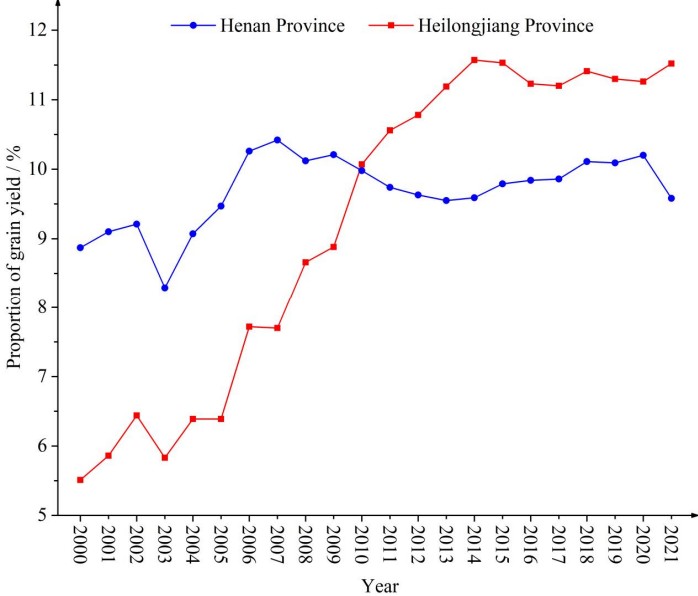

**Figure 5.** Changes in the proportion of grain yield in Henan Province and Heilongjiang Province from 2000 to 2021.

The regional disparities gradually decreased, and the development capacity became more balanced (Figure 6). The coefficient of variation in both provinces changed signif-

icantly. The coefficient of variation from Henan showed a slight increasing trend while Heilongjiang had a sharp decreasing trend, which further indicates the narrowing difference in capacity among counties in Heilongjiang. Henan consistently held the top position until 2010. The sown area of grain crops from 2000 to 2021 in Henan increased from 9.03 million hectares to 10.772 million hectares, with a decreasing difference and relatively stable regional changes. On the other hand, Heilongjiang has a relatively shorter history of agricultural development but has established modern and mechanized high-standard farmland demonstration bases. The sown area in Heilongjiang increased from 7.852 million hectares to 14.551 million hectares. Additionally, the fertile black land coupled with a longer growth cycle made for a higher grain yield per unit area. Thus, the coefficient of variation and county-level disparities gradual decreased in Heilongjiang.

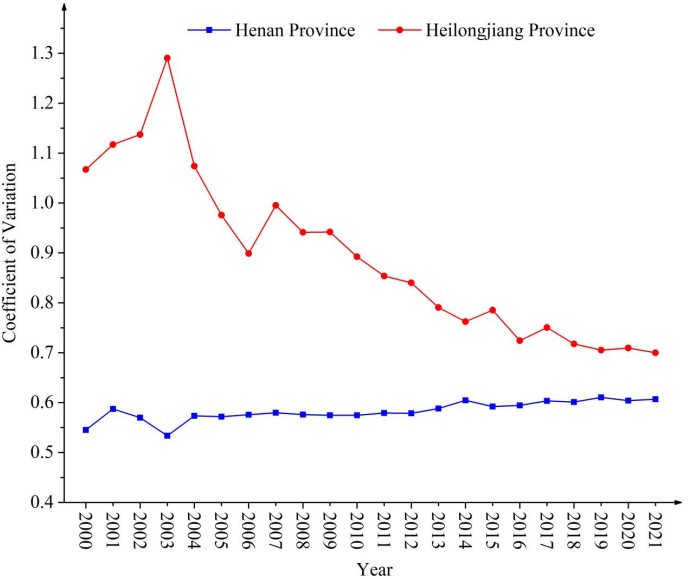

**Figure 6.** Coefficient of variation in grain yield in Henan and Heilongjiang Provinces from 2000 to 2021.

### 3.1.2. Spatial Variation Characteristics

The grain yield of all counties was set to five levels by using the natural breaks classification method, so as to obtain spatial distribution maps. The results of five specific years, 2000, 2005, 2010, 2015, and 2021, are shown in Figure 7.

(1) The number of high-yield counties increased and low-yield counties decreased. In 2000, only Hua County had more than an 800,000-ton yield, and 33 counties, mainly concentrated in the west, had less than 200,000 tons in Henan. By 2021, the number of counties with a weight exceeding 800,000 tons in the southeast of Henan increased to 27. There were only 16 counties with a weight below 200,000 tons, primarily located in western mountains. In 2000, there were six counties in Heilongjiang with more than 800,000 tons of yield, mainly distributed in the southwestern regions. Meanwhile, 26 counties had less than 200,000 tons. However, Heilongjiang showed an improvement from 2005 to 2021 and had 45 counties concentrated in the Songnen Plain with more than 800,000 tons by 2021. The number of counties with less than 200,000 tons decreased to 7. These counties were mainly in the northern high-latitude areas and southern mountainous regions.

(2) There was a differentiation between the high-yield and low-yield counties, with a concentrating trend in high-yield counties. Before 2010, the distribution in both provinces had a low concentration. However, there were a concentration of high-yield counties and a reduction in the spatial distribution range of low-yield counties after 2010. By 2021, high-yield counties in Henan were concentrated in the north, as well as in several cities such as Anyang and Shangqiu in the east, and the low-yield counties were mainly distributed in the west and northwestern mountainous region. In Heilongjiang, the high-yield counties

in 2021 more concentrated into two major high-yield areas in the west and east, while the center was in the Lesser Khingan Mountains and Changbai Mountains, forming a low-yield belt from the south to the north. The north was not suitable.

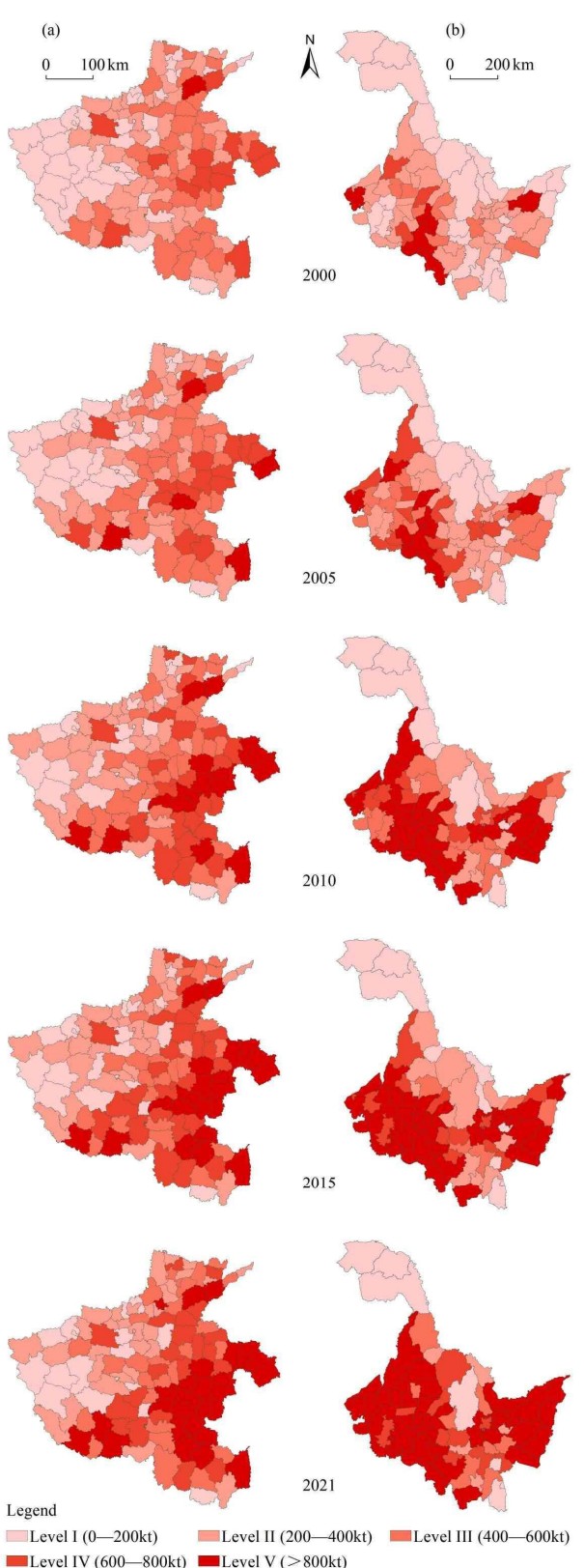

**Figure 7.** Spatial distribution of grain yield in counties.

### 3.1.3. Temporal and Spatial Evolution Trend

Standard deviation ellipses at the county level were generated by using a directional distribution tool (Figure 8). Based on the centroid coordinates and the ellipse range of the spatial distribution (Table 2), the spatiotemporal dynamic changes were further revealed.

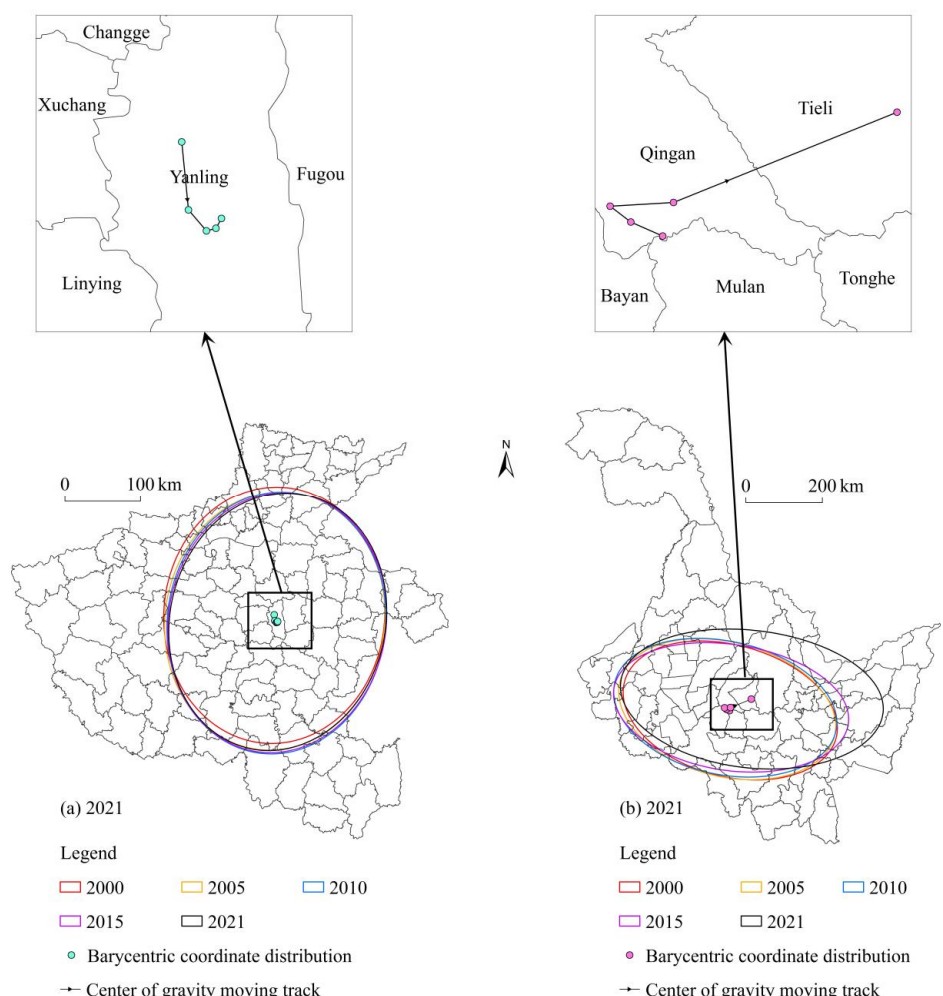

**Figure 8.** Spatial evolution of grain yield. Note: 2000, 2005, 2010, 2015, and 2021 represent the standard deviation ellipses of the corresponding years in Henan and Heilongjiang Provinces, respectively.

**Table 2.** Results of standard deviation ellipse calculation.

| District | Year | Standard Deviation Ellipse | | | | | |
|---|---|---|---|---|---|---|---|
| | | Area /km² | Longitude (°E) | Latitude (°N) | Long/km | Short/km | Azimuth (°) |
| Henan | 2000 | 78,475.55 | 114.15 | 34.06 | 146.48 | 170.54 | 2.71 |
| | 2005 | 78,627.02 | 114.16 | 33.99 | 145.51 | 172.00 | 6.95 |
| | 2010 | 79,322.38 | 114.18 | 33.97 | 144.73 | 174.46 | 5.43 |
| | 2015 | 78,655.41 | 114.19 | 33.97 | 144.32 | 173.49 | 8.12 |
| | 2021 | 77,341.12 | 114.20 | 33.98 | 142.82 | 172.38 | 11.54 |
| Heilongjiang | 2000 | 154,627.52 | 127.69 | 46.56 | 284.10 | 173.26 | 104.60 |
| | 2005 | 159,540.63 | 127.59 | 46.59 | 287.26 | 176.80 | 103.80 |
| | 2010 | 162,273.42 | 127.52 | 46.63 | 293.90 | 175.77 | 101.46 |
| | 2015 | 160,696.21 | 127.72 | 46.64 | 307.73 | 166.23 | 98.08 |
| | 2021 | 195,278.76 | 128.43 | 46.84 | 343.31 | 181.07 | 96.30 |

(1) The spatial distribution centroid changed obviously, where the center reflects the spatial centroid. From 2000 to 2021, the centroid in Henan initially moved noticeably towards the southeast and later slightly shifted towards the northeast, with the centroid consistently located within Xuchang City, Yanling County. In the same period, the centroid in Heilongjiang experienced a slight movement towards the northwest and then a significant shift towards the northeast.

(2) The spatial distribution exhibits the fluctuations in different regions, where the area of the ellipse reflects the range and the interior region represents the main area. The spatial distribution area in Henan had a decreasing trend, while in Heilongjiang, it had a significant upward trend. Compared to the year 2000, the spatial distribution area in Henan decreased by 1.45% in 2021, while Heilongjiang witnessed an increase of 26.29%.

(3) The orientation of the spatial distribution underwent a slight adjustment. From 2000 to 2021, the orientation angle in Henan gradually increased from 2.71° to 11.54°, indicating a clockwise rotation and a shift towards the northeast and southwest directions. The orientation angle in Heilongjiang gradually decreased from 104.60° to 96.30°, indicating a counterclockwise rotation.

The above changes indicate the dynamic evolution of spatial distribution. Considering the spatiotemporal characteristics, the spatial distribution in Henan had a southeast shifting trend, while it demonstrated a shift towards the northeast in Heilongjiang.

### 3.2. Spatial Correlation Analysis of Grain Yield

3.2.1. Spatial Autocorrelation Analysis

The Global Moran's *I* index was calculated using ArcGIS and GeoDa software to explore the spatial autocorrelation characteristics (Table 3). The spatial weight matrix was determined using the Queen contiguity matrix from the contiguity weight approach, which considered the neighbors based on the shared edges or vertices. A higher Moran's *I* value indicates a stronger spatial correlation, while a larger *Z* suggests a smaller average deviation. A smaller *P* indicates a more significant spatial correlation.

**Table 3.** Changes in income levels of low-income farmers.

| Year | Henan | | | Heilongjiang | | |
|---|---|---|---|---|---|---|
| | Moran's *I* | *Z* | *P* | Moran's *I* | *Z* | *P* |
| 2000 | 0.346 | 5.390 | 0.001 | 0.322 | 4.964 | 0.005 |
| 2001 | 0.418 | 6.420 | 0.001 | 0.280 | 4.419 | 0.004 |
| 2002 | 0.413 | 6.281 | 0.001 | 0.350 | 5.428 | 0.001 |
| 2003 | 0.221 | 3.521 | 0.002 | 0.379 | 5.986 | 0.004 |
| 2004 | 0.367 | 5.760 | 0.001 | 0.344 | 5.272 | 0.001 |
| 2005 | 0.361 | 5.648 | 0.001 | 0.348 | 5.293 | 0.001 |
| 2006 | 0.375 | 5.892 | 0.001 | 0.379 | 5.523 | 0.001 |
| 2007 | 0.382 | 5.994 | 0.001 | 0.392 | 5.737 | 0.001 |
| 2008 | 0.402 | 6.306 | 0.001 | 0.375 | 5.455 | 0.001 |
| 2009 | 0.421 | 6.589 | 0.001 | 0.361 | 5.300 | 0.001 |
| 2010 | 0.427 | 6.670 | 0.001 | 0.380 | 5.529 | 0.001 |
| 2011 | 0.440 | 6.860 | 0.001 | 0.355 | 5.127 | 0.001 |
| 2012 | 0.440 | 6.846 | 0.001 | 0.316 | 4.537 | 0.001 |
| 2013 | 0.456 | 7.115 | 0.001 | 0.343 | 4.951 | 0.001 |
| 2014 | 0.465 | 7.233 | 0.001 | 0.353 | 5.106 | 0.001 |
| 2015 | 0.439 | 6.848 | 0.001 | 0.391 | 5.553 | 0.001 |
| 2016 | 0.449 | 7.021 | 0.001 | 0.376 | 5.325 | 0.001 |
| 2017 | 0.466 | 7.360 | 0.001 | 0.380 | 5.442 | 0.001 |
| 2018 | 0.456 | 7.203 | 0.001 | 0.341 | 4.887 | 0.001 |
| 2019 | 0.444 | 6.906 | 0.001 | 0.405 | 5.675 | 0.001 |
| 2020 | 0.475 | 7.446 | 0.001 | 0.417 | 5.840 | 0.001 |
| 2021 | 0.438 | 6.955 | 0.001 | 0.415 | 5.801 | 0.001 |

The global spatial autocorrelation analysis revealed an increasing trend in the spatial autocorrelation. As shown in Table 3, the Global Moran's *I* values in Henan and Heilongjiang from 2000 to 2021 were both positive and significant at the 1% level, which indicated a significant positive spatial clustering, where counties with high (or low) yield were spatially adjacent. The Global Moran's *I* value exhibited three distinct phases in Henan: 2000–2004 (fluctuating phase), 2004–2014 (gradual increase phase), and 2014–2021 (fluctuating and alternating phase).

From 2000 to 2004, the Global Moran's *I* value had a fluctuating pattern resembling an "N" shape, with a maximum fluctuation range of 0.197. From 2004 to 2014, the Global Moran's *I* value gradually increased from 0.367 to 0.465. Then, the Global Moran's *I* value exhibited a fluctuating pattern resembling a "W" shape around 0.45. The Global Moran's *I* value in Heilongjiang had an overall upward fluctuating trend, with a fluctuation range of 0.137. Overall, there was an increasing trend in both Henan and Heilongjiang Provinces. The growth rate of the Global Moran's *I* values for grain yield in Henan was higher than that in Heilongjiang.

3.2.2. Local Spatial Autocorrelation Analysis

Local spatial autocorrelation analysis was conducted to examine the county-level grain yield in different periods (Figure 9). The H-H cluster represents the counties with a high yield and surrounded by neighboring counties with a high yield as well. The L-L cluster represents the counties with a low yield and surrounded by neighboring counties with a low yield. The H-L cluster represents the counties with a high yield and surrounded by neighboring counties with a low yield, while the L-H cluster represents the counties with a low yield surrounded by neighboring counties with a high yield. Overall, the gradually increasing "H-H" and "L-L" clusters were the most widely distributed patterns in both provinces.

Figure 9 shows that, in Henan, the number of H-H clusters increased from 8 to 14 and the number of L-L clusters increased from 12 to 23. The number of H-L and L-H clusters was relatively stable, but their distribution significantly varied in the different regions. Overall, the number of non-significant areas gradually decreased. Regionally, the H-H clustering areas in Henan were concentrated in Shangqiu City, Zhoukou City, and Zhumadian City. These regions served as core areas with high yields. Additionally, the H-H clustering areas gradually spread to the southeastern part of Henan. The L-L clustering areas in Henan were concentrated and contiguous in Zhengzhou City, Luoyang City, Sanmenxia City, Pingdingshan City, and Jiaozuo City. Thus, these areas were unsuitable for large-scale agricultural production.

With a significant change in regional distribution, the number of clustering types in Heilongjiang remained stable. The H-H clustering areas were more concentrated and exhibited more notable movement. From 2000 to 2015, the H-H clustering areas were mainly concentrated in the southwestern region and gradually increased. These areas were located on the Songnen Plain, where the closed rivers ensured abundant water resources. By 2021, the H-H clustering areas shifted to the easternmost region, known as the Sanjiang Plain. The L-L clustering areas were distributed in the northern high-latitude region and the southeastern Changbai Mountains area. In the southeastern Changbai Mountains area, the limited agricultural production led to L-L clustering areas.

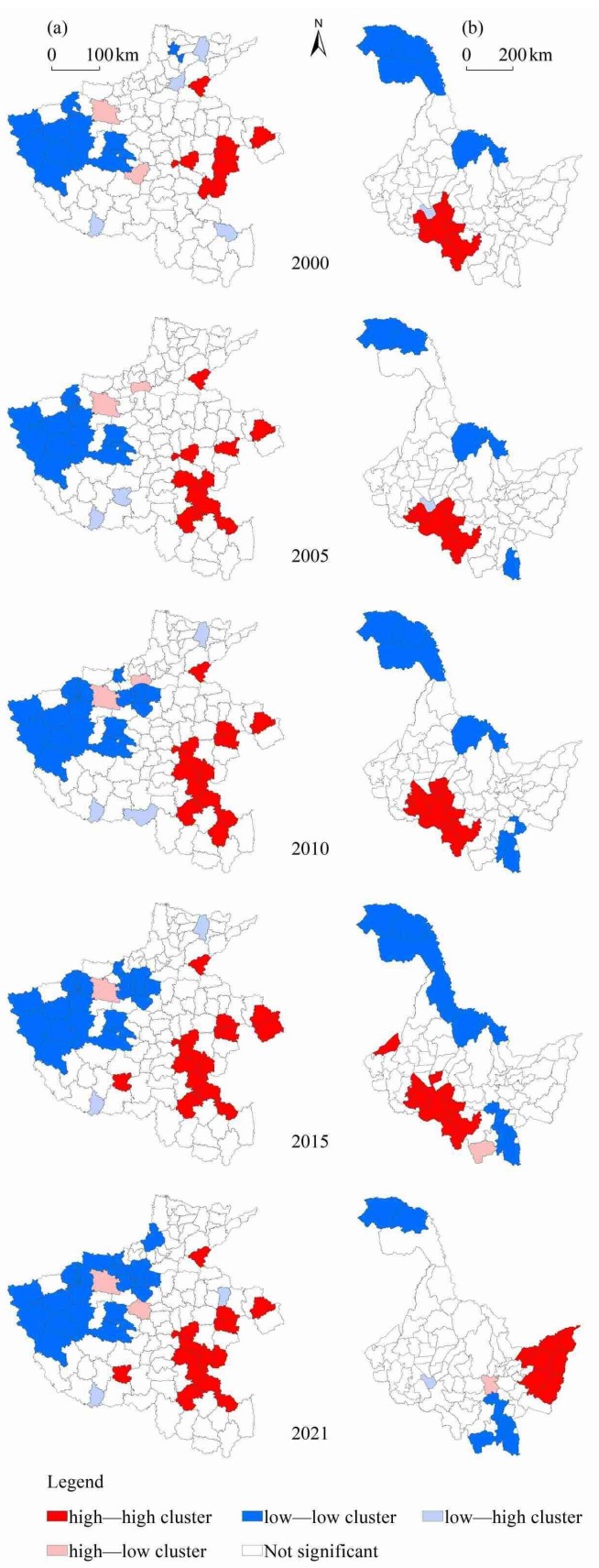

**Figure 9.** Spatial clustering differences in grain yield based on LISA Aaalysis.

### 3.3. Analysis of Influencing Factors

In order to identify the key factors and explore their strength and direction, 12 relevant factors across 3 dimensions were analyzed by employing econometric and geographical methods [82–84].

### 3.3.1. Geographical Detector Analysis

Firstly, a geographical detector was used to explore the 12 indicators (Figure 10). The 3 categories of geographical environment, socio-economic factors, and input factors had different effects in both provinces. The input factors had the strongest impact, followed by the geographical environment, while the socio-economic factors had the weakest influence. The main factor in Henan was the area of grain crop planting ($x_9$), with a high impact intensity of 0.768. The total power of agricultural machinery ($x_{10}$) was the next significant factor, with an impact intensity of 0.549. On the other hand, the influence of the proportion of agricultural labor to the total population ($x_5$) was the weakest, with an impact intensity of only 0.054. Similarly, in Heilongjiang, the primary influencing factors were also the area of grain crop planting ($x_9$) and the total power of agricultural machinery ($x_{10}$), with impact intensities of 0.593 and 0.464, respectively. However, the impact intensity values were lower than those in Henan. In Heilongjiang, the proportion of agricultural labor to the total population ($x_5$), annual precipitation ($x_2$), the proportion of primary industry output value to GDP ($x_7$), and annual sunshine hours ($x_3$) had minimal impacts, and their impact intensities were not significant.

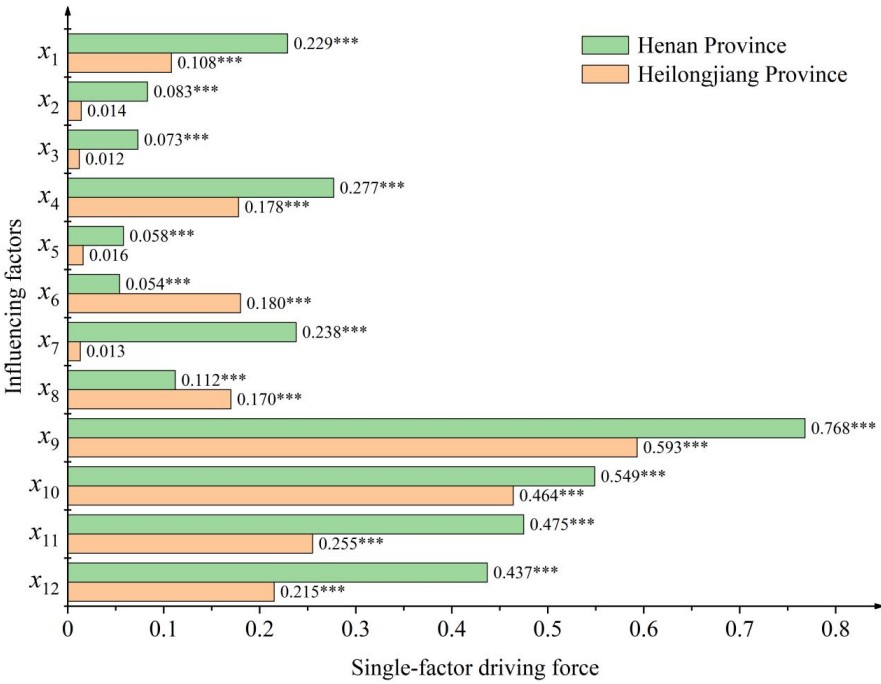

**Figure 10.** Results of factor detection. Note: *** indicates that the q-value is significant at the 1% level.

### 3.3.2. Geographically Weighted Regression

A geographical weighted regression was applied to further explore the effects of each variable, with overall and local regression results.

(1) Overall Regression Results.

Before the geographically weighted regression, a general linear regression analysis based on the ordinary least squares (OLS) was performed to examine the influencing factors. The following steps were taken: (1) standardization of the original data to eliminate the impact of different scales; (2) stepwise regression to select important factors that were both significant and free from multicollinearity. Finally, six significant factors were identified

for each province through significance tests and collinearity diagnostics, as shown in Tables 4 and 5.

**Table 4.** OLS regression results for Henan Province.

| Dependent Variable | 2000 | | 2005 | | 2010 | | 2015 | | 2021 | |
|---|---|---|---|---|---|---|---|---|---|---|
| | Regression Coefficient | VIF | R | VIF | Regression Coefficient | VIF | Regression Coefficient | VIF | Regression Coefficient | VIF |
| $x_3$ | 0.133 *** | 1.587 | 0.064 *** | 1.353 | 0.042 ** | 1.132 | 0.031 * | 1.375 | 0.021 * | 1.183 |
| $x_4$ | −0.084 ** | 1.620 | −0.068 *** | 1.621 | −0.131 *** | 1.331 | −0.124 *** | 1.421 | −0.117 *** | 1.356 |
| $x_9$ | 0.718 *** | 4.588 | 0.787 *** | 4.966 | 0.887 *** | 4.340 | 0.878 *** | 4.967 | 0.792 *** | 6.624 |
| $x_{10}$ | 0.107 ** | 2.612 | −0.028 * | 2.949 | −0.093 ** | 3.528 | −0.072 * | 4.168 | −0.134 *** | 4.329 |
| $x_{11}$ | −0.059 * | 4.265 | 0.002 * | 3.137 | 0.042 * | 3.316 | 0.060 * | 3.264 | 0.233 *** | 3.639 |
| $x_{12}$ | 0.263 *** | 3.446 | 0.224 *** | 4.106 | 0.098 *** | 1.998 | 0.092 *** | 1.952 | 0.037 * | 1.886 |
| F | 183.293 *** | | 350.860 *** | | 294.624 *** | | 422.691 *** | | 257.129 *** | |
| Koenker | 10.899 * | | 16.309 ** | | 14.910 ** | | 18.341 *** | | 13.950 ** | |
| AICc | 72.046 | | −0.407 | | 19.394 | | −21.714 | | 34.682 | |
| $R^2$ | 0.907 | | 0.949 | | 0.940 | | 0.957 | | 0.932 | |
| Adj $R^2$ | 0.902 | | 0.946 | | 0.937 | | 0.955 | | 0.928 | |

Note: *, **, and *** represent statistical significance at the 10%, 5%, and 1% levels, respectively.

**Table 5.** OLS regression results for Heilongjiang Province.

| Dependent Variable | 2000 | | 2005 | | 2010 | | 2015 | | 2021 | |
|---|---|---|---|---|---|---|---|---|---|---|
| | Regression Coefficient | VIF | Regression Coefficient | VIF | Regression Coefficient | VIF | Regression Coefficient | VIF | Regression Coefficient | VIF |
| $x_1$ | −0.045 * | 1.252 | 0.039 * | 1.230 | 0.107 ** | 1.286 | 0.153 *** | 1.274 | 0.137 ** | 1.263 |
| $x_8$ | −0.015 * | 1.117 | 0.104 *** | 1.099 | 0.058 * | 1.061 | 0.023 * | 1.024 | −0.086 *** | 1.135 |
| $x_9$ | 0.130 * | 3.370 | 0.294 *** | 4.102 | 0.271 *** | 3.554 | 0.213 ** | 3.394 | 0.683 *** | 1.897 |
| $x_{10}$ | 0.330 *** | 4.110 | 0.003 * | 4.003 | 0.064 * | 4.382 | 0.206 ** | 4.020 | 0.150 ** | 2.085 |
| $x_{11}$ | 0.485 *** | 4.918 | 0.650 *** | 4.631 | 0.558 *** | 4.149 | 0.415 *** | 3.493 | 0.036 * | 1.085 |
| $x_{12}$ | 0.106 ** | 1.473 | 0.056 * | 1.530 | 0.064 * | 1.734 | 0.125 ** | 1.909 | 0.205 *** | 1.677 |
| F | 81.332 *** | | 157.552 *** | | 73.562 *** | | 70.794 *** | | 102.005 *** | |
| Koenker | 22.030 *** | | 12.057 * | | 12.120 * | | 15.713 ** | | 33.514 *** | |
| AICc | 72.422 | | 27.406 | | 78.991 | | 81.476 | | 57.319 | |
| $R^2$ | 0.878 | | 0.933 | | 0.867 | | 0.862 | | 0.900 | |
| Adj $R^2$ | 0.867 | | 0.927 | | 0.855 | | 0.850 | | 0.891 | |

Note: *, **, and *** represent statistical significance at the 10%, 5%, and 1% levels, respectively.

Table 4 shows that the R-squared values in Henan for the years 2000, 2005, 2010, 2015, and 2021 are 0.907, 0.949, 0.940, 0.957, and 0.932, respectively. The corresponding adjusted R-squared values are 0.902, 0.946, 0.937, 0.955, and 0.928. The F-values and $p$-values for all variables are statistically significant at the 1% level. Additionally, the variance inflation factor (VIF) values for all variables are less than 7.5. Among the factors, the sown area of grain crops, effective irrigated area, and average annual temperature had a positive effect, while average slope had a negative effect. The total power of agricultural machinery had a positive effect on in the year 2000 but had a negative effect in 2005, 2010, 2015, and 2021. Similarly, the net application number of chemical fertilizers had a negative effect in 2000 but had a positive effect in 2005, 2010, 2015, and 2021.

According to Table 5, the R-squared values for the OLS models in Heilongjiang for the years 2000, 2005, 2010, 2015, and 2021 are 0.878, 0.933, 0.867, 0.862, and 0.900, respectively. The corresponding adjusted R-squared values are 0.867, 0.927, 0.855, 0.850, and 0.891. The F-values and p-values for all variables are also significant at the 1% level. The VIF values for all variables are less than 7.5, indicating a good fit. Factors such as the sown area of grain crops, total power of agricultural machinery, net application number of chemical fertilizers, and effective irrigated area had a positive effect. The average annual temperature had a

negative effect in 2000 but had a positive effect in the years 2005, 2010, 2015, and 2021. The per capita net income of farmers had a negative effect in the years 2000 and 2021 but had a positive effect in the years 2005, 2010, and 2015.

(2) Local Regression Results.

The GWR regression was conducted on the six influential factors that passed the significance test and collinearity diagnosis. Tables 6 and 7 present the results of the model parameters for the influencing factors in Henan and Heilongjiang in 2000, 2005, 2010, 2015, and 2021, where the AICc values of GWR were lower than those of OLS and the R-squared and adjusted R-squared values were higher in GWR compared to OLS.

**Table 6.** GWR results for Henan Province.

|  | **2000** | **2005** | **2010** | **2015** | **2021** |
|---|---|---|---|---|---|
| Bandwidth | 202,826.487 | 247,709.211 | 133,818.278 | 106,874.926 | 190,312.857 |
| Residual Squares | 8.659 | 5.117 | 3.065 | 1.571 | 6.303 |
| Effective Number | 19.256 | 15.464 | 33.106 | 43.728 | 20.830 |
| Sigma | 0.293 | 0.221 | 0.188 | 0.144 | 0.252 |
| AICc | 61.716 | −8.297 | −31.641 | −79.357 | 26.825 |
| $R^2$ | 0.927 | 0.957 | 0.974 | 0.987 | 0.947 |
| Adjusted $R^2$ | 0.914 | 0.951 | 0.965 | 0.979 | 0.936 |

**Table 7.** GWR results for Heilongjiang Province.

|  | **2000** | **2005** | **2010** | **2015** | **2021** |
|---|---|---|---|---|---|
| Bandwidth | 3.163 | 13.813 | 3.848 | 2.933 | 2.884 |
| Residual Squares | 5.393 | 4.842 | 7.122 | 5.097 | 2.059 |
| Effective Number | 20.881 | 8.260 | 18.178 | 24.233 | 22.821 |
| Sigma | 0.316 | 0.269 | 0.354 | 0.317 | 0.199 |
| AICc | 62.316 | 27.233 | 76.019 | 67.577 | −4.303 |
| $R^2$ | 0.927 | 0.935 | 0.904 | 0.931 | 0.972 |
| Adjusted $R^2$ | 0.900 | 0.927 | 0.875 | 0.900 | 0.961 |

(3) Spatial Heterogeneity Analysis of Influential Factors.

To further investigate the local spatial characteristics impacted by the six significant variables, spatial heterogeneity analysis of influential factors was conducted based on the regression results of GWR (Figures 11–14).

(1) The Influence of Sunshine Duration in Henan Province.

In 2000 and 2005, the impact of annual sunshine hours was relatively similar, gradually transitioning from the high-value areas to the low-value areas from south to north; In 2010, the intensity of the action gradually weakened from the north to the south, with the highest values concentrated in the north and southeast. In 2015, the high-value areas were in the southeast, the central region mostly had a moderate intensity, and Nanyang in the southwest had a weaker intensity. By 2021, Nanyang had the strongest impact, and Henan formed a distribution of decreasing intensity from southwest to northeast.

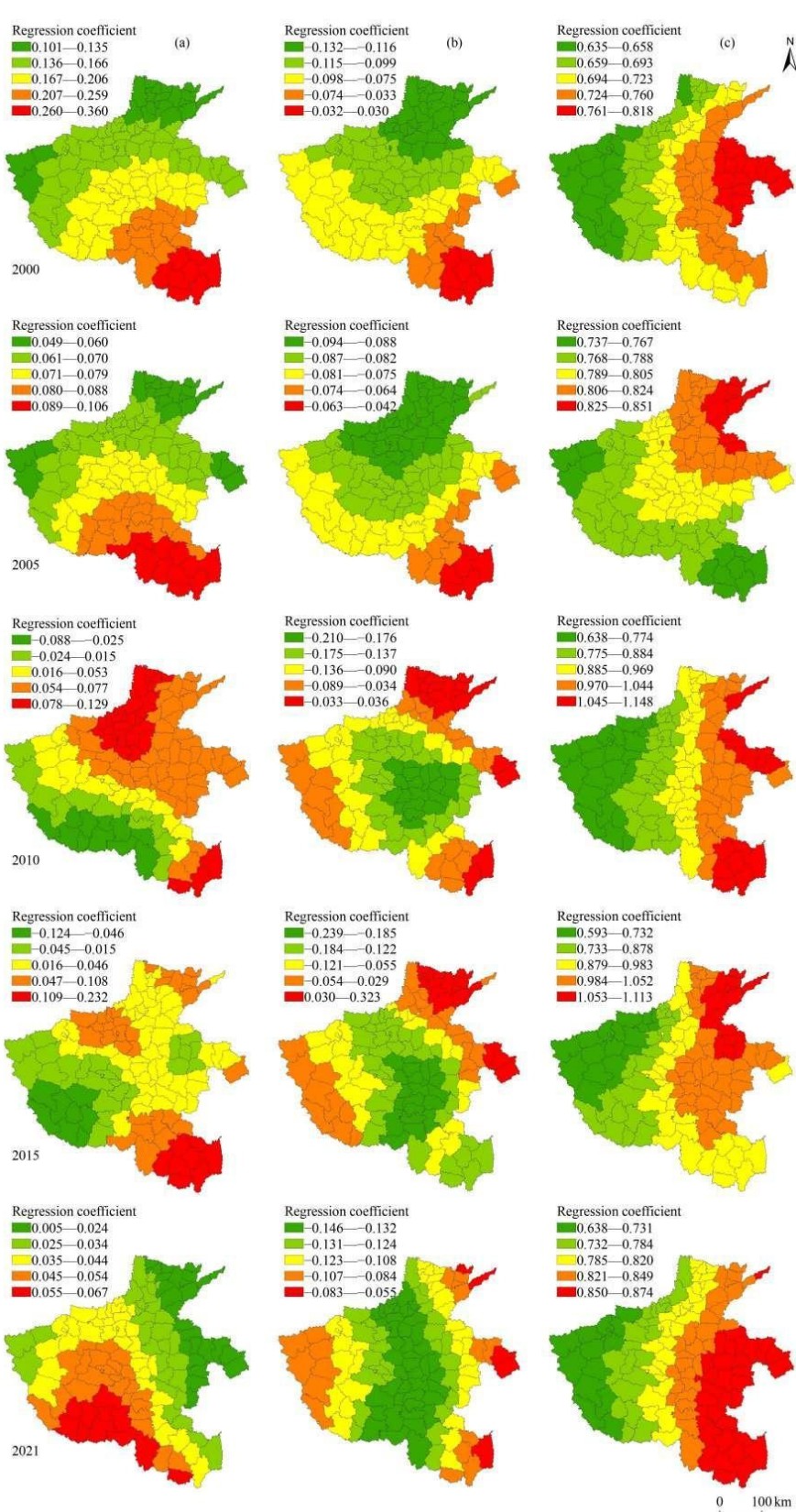

**Figure 11.** Spatial distribution of regression coefficients for influential factors on grain yield in Henan Province (**a–c**). Note: Henan Province's (**a**), (**b**), and (**c**), respectively, represent annual sunshine duration, average slope, and sown area of grain crops.

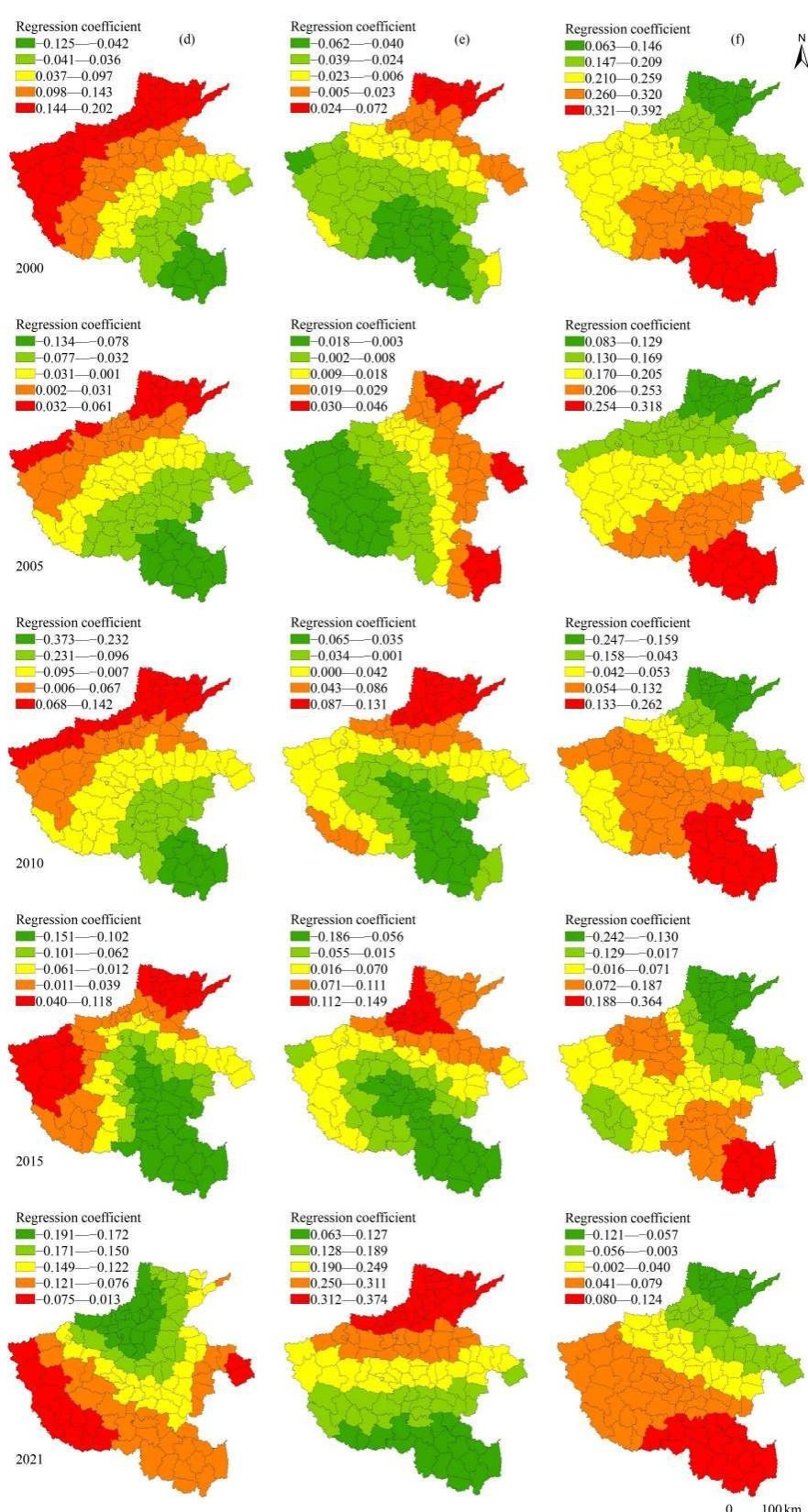

**Figure 12.** Spatial distribution of regression coefficients for influential factors on grain yield in Henan Province (**d**–**f**). Note: Henan Province's (**d**), (**e**), and (**f**), respectively, represent total agricultural machinery power, pure amount of fertilizer applied, and effective irrigation area.

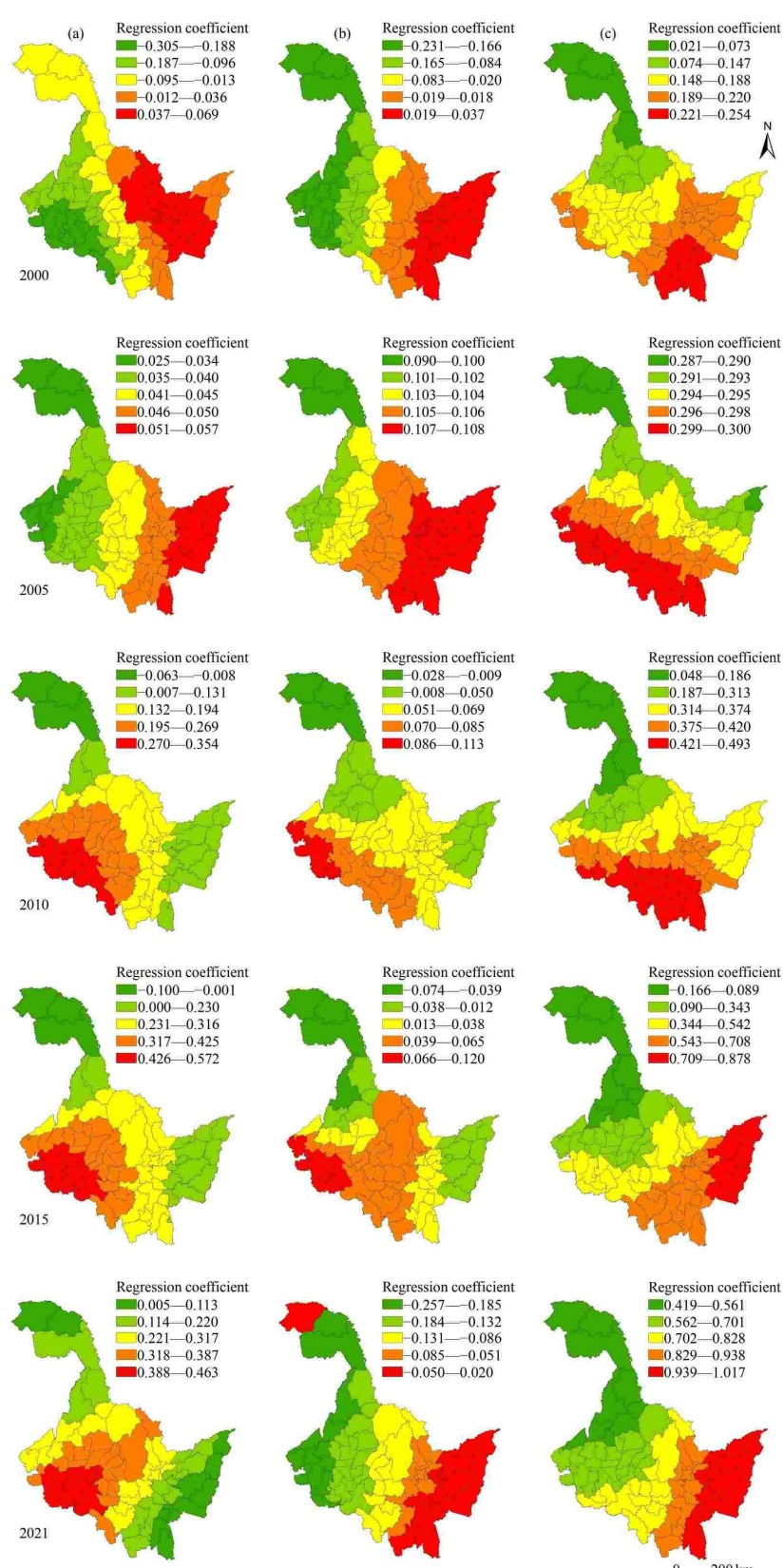

**Figure 13.** Spatial distribution of regression coefficients for influential factors on grain yield in Heilongjiang Province (**a**–**c**). Note: Heilongjiang Province's (**a**), (**b**), and (**c**), respectively, represent average annual temperature, per capita net income of farmers, and sown area of grain crops.

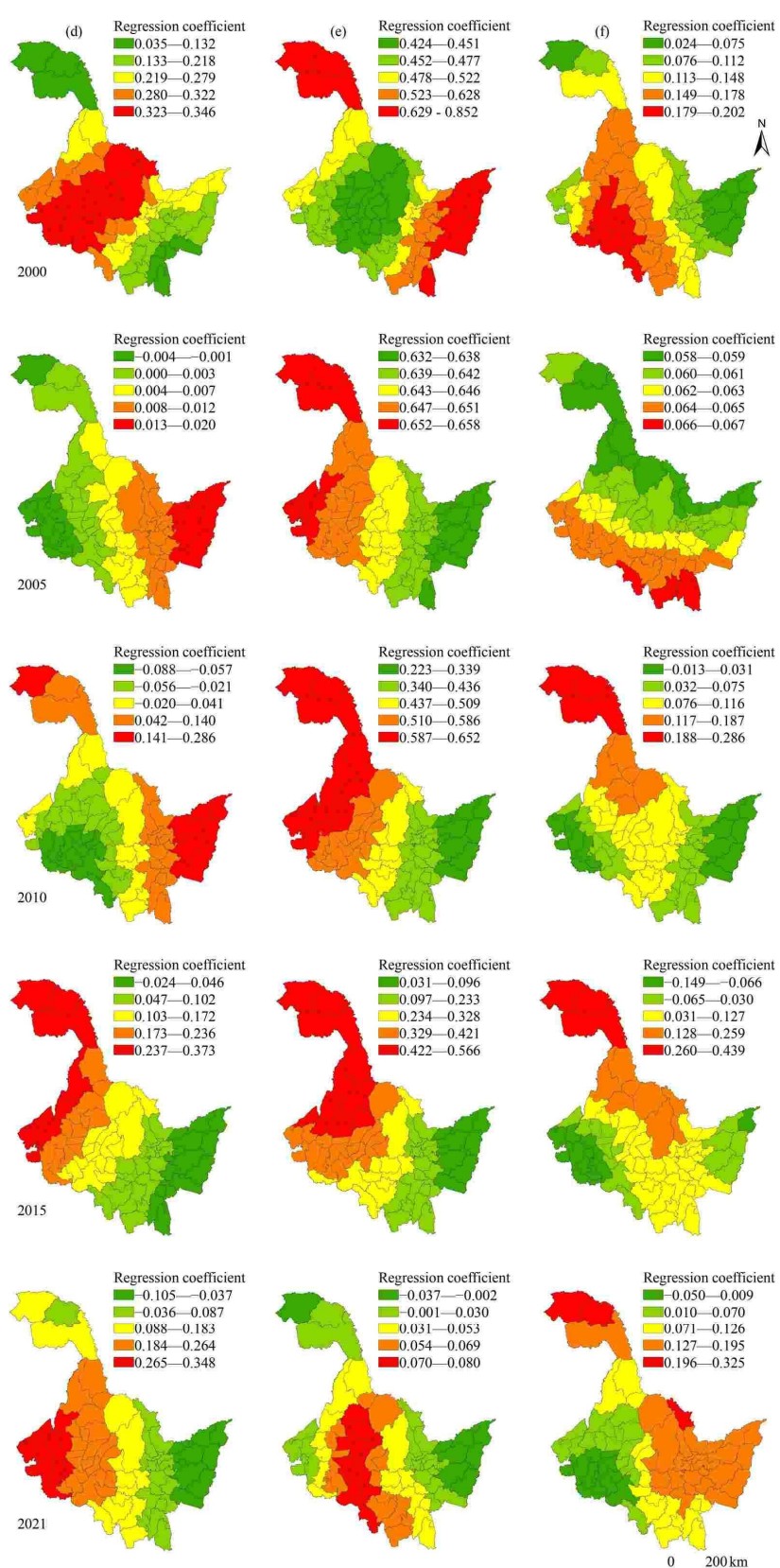

**Figure 14.** Spatial distribution of regression coefficients for influential factors on grain yield in Heilongjiang Province (**d**–**f**). Note: Heilongjiang Province's (**d**), (**e**), and (**f**), respectively, represent total agricultural machinery power, pure amount of fertilizer applied, and effective irrigation area.

(2) The Influence of Average Slope in Henan Province.

The intensity of the average slope was almost always negative, and there was only a positive effect in some regions in 2010 and 2015, but the effect varied significantly among different regions. In 2000 and 2005, the intensity of the action gradually increased from the north to the south. In 2010, 2015, and 2021, the trend was characterized by a gradual increase in the intensity of outward action from the central region, with the weaker areas gradually increasing and the intensity value gradually decreasing, indicating that the impact of average slope was gradually weakening.

(3) The Influence of Average Temperature in Heilongjiang Province.

In 2000 and 2005, the intensity of the average annual temperature gradually weakened from the east to the west, with the highest value appearing in the Sanjiang Plain area in the east, while the low-value areas were in the Songnen Plain and Great Xing'an Mountains Forest in the west. In 2010, 2015, and 2021, there was a gradual weakening trend from the southwest to the east and north. The Songnen Plain became a high-value area, while the Sanjiang Plain, Changbai Mountain, and higher-latitude areas in the north were the low-value areas. Overall, the impact of average annual temperature showed a gradual increase from 2000 to 2021, but the intensity varied significantly.

(4) The Influence of Per Capita Net Income of Farmers in Heilongjiang Province.

From 2000 to 2021, the intensity of the per capita net income of farmers also showed a trend of first increasing and then decreasing. In 2000, 2005, and 2021, the spatial distribution had a decreasing trend from the Sanjiang Plain in the east to the west, and the intensity of the action gradually decreased. In 2010 and 2015, the high-value areas were distributed in the Songnen Plain area in the west, gradually weakening from the southwest to the east and north.

(5) The Influence of the Sown Area of Grain Crops in Both Provinces.

The coefficient value of the sown area of grain crops in Henan remained above 0.6 as a whole. In 2000, 2005, 2010, and 2015, the regression coefficient gradually increased, but decreased significantly by 2021. The yield of areas with large sown lands failed to increase correspondingly. Therefore, the regression coefficient decreased. The regions with high regression coefficients were generally distributed on the plains of the central and eastern regions, while the western regions were relatively low. The plains in the central and eastern regions had a vast area and a large planting area and became the core areas with high regression coefficients. The western region, mountainous and hilly, had limited land areas.

The coefficient values in Heilongjiang remained relatively low in 2000, 2005, and 2010, with the highest value lower than 0.5. However, by 2015 and 2021, the overall regression coefficient increased largely and ranked the first. The distribution of low-value areas was relatively stable, mainly located in the high-latitude and high-altitude areas in the northwest. In 2000 and 2010, the regression coefficient showed a latitudinal belt from the south to the north. In 2005, there was a decrease from southwest to northeast. By 2015 and 2021, there was a decrease from southeast to northwest. The high-value areas were distributed on the eastern Sanjiang Plain, concentrated in contiguous areas, and gradually expanded.

(6) The Influence of Agricultural Machinery Power in Both Provinces.

In Henan, the high-value areas of agricultural machinery total power had a transformation from the northwest to the southwest and southeast. In 2000, 2005, and 2010, the high-value areas were concentrated in the northwest region, with a decreasing trend from northwest to southeast, and the lowest value appeared in Xinyang in the southeast. Then, the high-value areas were gradually narrowing. In 2015, there was a significant change, which was concentrated in the western and northern regions, while the central and southern regions became low-value areas. By 2021, the regional distribution changed, with the high-value areas concentrated in the southwest and west, while they were scattered in Shangqiu in the east and there were low-value areas the in mountainous and hilly parts of the central north.

The coefficient in Heilongjiang had an overall trend of decreasing first and then increasing. In 2000, the high-value area was the largest and widely distributed in the southwest and central regions. As a result, the regression coefficient gradually decreased towards the southeast and northwest. In 2005 and 2010, the regression coefficient was the largest on the eastern Sanjiang Plain, gradually decreasing towards the west. Compared to 2005, the regression coefficient in 2010 significantly increased, and high-value areas also appeared in the northernmost city of Mohe. In 2015, the spatial distribution was the opposite of 2005, where the highest coefficient was in the northwest, decreasing from northwest to southeast. By 2021, the distribution was a west–east decreased belt-pattern.

(7) The Influence of Fertilizer Application Rate in Both Provinces.

The distribution impact of fertilizer application in Henan was relatively stable, where the high-value areas were generally distributed in the north and east. In 2000, 2010, 2015, and 2021, the strong-effect regions were in the northern cities of Anyang, Puyang, Hebi, and Xinxiang. The overall spatial pattern showed a gradual decrease from the north to the south, with the lowest values in the center and south. Xinyang had a small value, indicating a low impact of fertilizer application. In 2005, the distribution was significantly different, with a decreasing trend from the east to the west. The highest values were in Anyang and Puyang of the north, Shangqiu of the east, and Xinyang of the south. Low-value areas were in mountainous and hilly parts of the west.

In 2000 and 2021, the regression coefficients showed the opposite spatial distribution pattern in Heilongjiang. In 2000, the high-value areas were mainly on the eastern Sanjiang Plain and in the north, while the central area was low overall. By 2021, the high-value areas were mainly in the central area, while Sanjiang Plain became a low-value area. In 2005, 2010, and 2015, the high-value areas were distributed in the western Greater Khingan Mountains and the northern areas. The regression coefficient showed a large decrease in 2021, indicating a very weak impact.

(8) The Impact of Effective Irrigated Area in Both Provinces.

In Henan, the regression coefficient of effective irrigation area was the opposite of fertilizer application. The high-value areas were distributed in Xinyang City and Zhumadian City of the south, while the low-value areas were in the northern region. The secondary high-value areas gradually moved towards the western region with an increasing number of counties.

The regression coefficient In Heilongjiang first decreased and then gradually increased at five-time interfaces. In 2000, the high-value areas were mainly on the Songnen Plain of the southwest, while the low-value areas were in the north and on the Sanjiang Plain in the east. In 2005, the regression coefficient decreased significantly and the spatial distribution also decreased from the south to the north. In 2010 and 2015, the high-value area changed into the north and most central areas, while the western Songnen Plain and the eastern Sanjiang Plain were relatively low. By 2021, Sanjiang Plain changed from a low-value to a high-value area.

## 4. Discussion

This paper mainly explored the differences in grain yield changes and influencing factors, comparing those between Henan and Heilongjiang Provinces in China. Firstly, the variation coefficients, standard deviation ellipses, and ESDA methods were used to compare and analyze the characteristics of grain yield in the two provinces. Then, by combining geographic detectors with geographic weighted regression methods, the mechanism of the main influencing factors was further elaborated. The research results indicate that the spatiotemporal changes in grain yield in the two provinces were significant, and the three-dimensional influencing factors also played a significant role. The variation characteristics in the two provinces were more representative at the county level, and the influencing factors also showed strong regional differences. Based on the above, certain discussions are prompted:

### 4.1. Spatiotemporal Characteristics of Grain Yield

1.  From the perspective of temporal changes, the total grain yield in Henan steadily increased, with only minor fluctuations, but the annual average growth rate and amount remained stable, which was similar to the traditional agricultural provinces of Shandong and Hunan in China [14,60]. The grain yield of Heilongjiang increased sharply, making it the province with a largest grain yield and a high annual growth rate, which was similar to the trend of Jilin and Liaoning Provinces in Northeast China [45]. In recent years, these regions experienced a drastic increase in grain yield, which played a prominent role in food security.

2.  There were some significant county-level differences. The grain yield in Henan generally showed a gradual increase from the west to the east, which was consistent with the overall distribution in China [15]. Many previous studies of China also showed this distribution pattern [49], so this may be a common phenomenon in most provinces of China. The grain yield of counties in Heilongjiang was relatively low in the eastern and western regions, with a high yield in the central and northern regions. The result was different from the distribution of most other provinces in China, demonstrating the unique spatial distribution of grain yield.

### 4.2. Influencing Factors of Grain Yield

1.  Factor input is the primary guarantee for grain yield. According to the results of geographic detectors and geographic weighted regression, the sown area of grain crops, the total power of agricultural machinery, the reduced net amount of fertilizer application, and the effective irrigation area all had a significant impact on the grain yield of the two provinces, which was consistent with the previous results [85–90]. Therefore, the primary prerequisite is to limit the non-agricultural and non-grain conversion of farmland, ensuring the sown area of grain crops. In addition, the investment in agricultural machinery, fertilizers, and irrigation was also important. The continuous popularization of agricultural machinery has not only reduced the cost of labor but also enabled intensive cultivation. Fertilizer can increase the grain yield by improving the required nutrients. The agricultural areas in China were mainly located in the monsoon areas, but the unstable precipitation left them prone to periodic droughts. In addition, improving the effective irrigation rate in agriculture was also a key factor in the increasing grain yield.

2.  The geographical environment is the foundation for the growth of food crops. The average annual temperature, annual precipitation, annual sunshine hours, and average slope also had a significant impact on the grain yield of the two provinces, which is a similar finding to existing research results [35–37,91]. A lower average annual temperature can make for a shorter growth period and frost, which has an adverse effect. Thus, the success of agriculture depends on the effective use of precipitation in irrigated farming areas, and the sunlight hours also affect the growth and quality of food crops by affecting plant photosynthesis.

3.  Socio-economics are an important driving force. The impact of per capita net income of farmers in Heilongjiang on grain yield was quite prominent, which was similar to the previous research results [31,32]. In Heilongjiang, the advantage of fewer people and more land led to a larger per capita arable land area, and rural laborers often choose to obtain a greater agricultural income. Therefore, the grain harvest and price rise can increase the agricultural income and enthusiasm, which further increase the sown area of grain crops, working to ensure national food security.

### 4.3. Recommendations

At present, the grain yield capacity is important to help ensure food security, whose influencing factors are complex and diverse. Thus, it is necessary for the government to develop tailored incentives based on the regional agricultural development. Additionally, the academic community should conduct innovative research to explore the processes,

patterns, and mechanisms of grain yield. Based on these considerations, in this paper, we make the following recommendations:

1.  Ensure the input of factors in the grain yield process. The planting area of grain crops has the greatest impact on grain yield, so a stable planting area is a key to ensuring grain yield. At the same time, the main focuses should be on increasing grain yield per unit area, introducing or cultivating high-quality grain varieties, promoting high-yield and high-density varieties, and utilizing agricultural technology to improve grain yield per unit area, especially in the large agricultural regions such as Henan and Heilongjiang. We need to increase the promotion of agricultural machinery, improve agricultural production efficiency, and reduce production costs. In addition, we can strengthen the construction of farmland water conservancy facilities that enhance the ability of the two provinces to resist drought and flood risks in agriculture, accelerate the development of high-standard farmland, strictly prevent the "cutting of green crops to destroy grain" of grain crops, and resolutely curb the "nonagricultural" farmland and "non-grain" agriculture.

2.  Promote agricultural modernization, intelligence, and digitization to develop smart agriculture. We need to strengthen meteorological monitoring of agricultural production and reduce the adverse effects of extreme weather on the growth of food crops. At the same time, we should strengthen the top-level design of smart agriculture, increase monitoring of agricultural diseases and pests, reduce labor costs through aerial spraying of pesticides and fertilizers, and improve intelligent control during the growth process of food crops. In addition, we can strengthen a series of services after grain production, build grain-drying facilities, provide grain storage facilities, solve sales difficulties, reduce the risks of food production in both pre- and post-production stages, and promote high-quality agricultural development.

3.  Ensure the income of grain farmers and mobilize their enthusiasm, reduce the cost of grain yield, and improve the mechanism for ensuring the income of grain farmers and compensating for the interests of main production areas. The government can regularly conduct outreach and training on agricultural planting, providing services such as agricultural supply, plant protection, and prevention, and improve planting skills and management capabilities. In addition, we can encourage various regions to develop the grain-processing industry according to local conditions, extend the industrial chain of agricultural production, promote the transformation and value-added of agricultural products, tap into the internal income potential of the grain industry, and fully mobilize the enthusiasm of farmers to increase the grain yield.

## 5. Conclusions

This study focused on the county-level grain yield in Henan Province and Heilongjiang Province, investigating the spatiotemporal evolution of grain yield from 2000 to 2021 by using methods such as coefficient of variation, standard deviation ellipse, and ESDA analysis. Furthermore, the driving factors behind the spatial variations were analyzed by utilizing geographic detectors and geographically weighted regression models. The conclusions are as follows:

*   There were significant differences in grain yield, annual growth rate, proportion in the country, and county-level production. The grain yield capacity of the two provinces gradually increased, and the annual average growth rate was higher than that of the whole country. The growth rate of the proportion of countrywide grain yield in Heilongjiang was greater than that in Henan. The county-level changes in grain yield of Henan remained stable, while the county-level differences in Heilongjiang gradually narrowed. The number of high-yield counties for grain yield significantly increased, and the spatial distribution of high-yield areas tends to be concentrated. The main areas of grain yield shifted significantly. The spatial distribution center of grain yield in Henan moved relatively little, while the distribution center of grain yield in Heilongjiang shifted significantly.

- The spatial correlation of grain yield was gradually increasing. The degree of spatial agglomeration was gradually strengthening, and the Global Moran's index increased. Moran's *I* value in Henan increased by a greater margin than that in Heilongjiang. Local spatial autocorrelation was mainly characterized by L-L and H-H clustering, and the number of counties in both types of clustering significantly increased. In addition, the spatial distribution range of H-H clustering areas in Heilongjiang changed significantly. The overall spatial pattern of grain yield in counties in Henan was stable, while the spatial pattern in Heilongjiang significantly changed.
- There were significant differences in the influencing factors of grain yield. Geographical environment, socio-economic factors, and factor inputs all had an impact on grain yield. Factor inputs had the greatest impact on grain yield, followed by geographical environment factors, and socio-economic factors had the smallest impact. The overall intensity of various factors in Henan Province was greater than that in Heilongjiang.
- The regional differences in the intensity of each influencing factor were significant. The intensity and direction of different influencing factors varied significantly. The sown area of grain crops, total power of agricultural machinery, reduced net amount of fertilizer application, and effective irrigation area all had a significant impact on grain yield, which was common to the two provinces. The annual sunshine hours and average slope had a significant impact in Henan, while the average annual temperature and per capita net income of farmers had a significant impact in Heilongjiang, which was a differential factor between the two provinces.

**Author Contributions:** Conceptualization, G.C. and Z.W.; methodology, Z.W.; software, Z.W. and G.C.; validation, G.C. and Z.W.; formal analysis, E.Z. and Z.W.; investigation, Z.W.; resources, Z.W.; data curation, Z.W.; writing—original draft preparation, Z.W.; writing—review and editing, G.C. and Z.W.; visualization, Z.W. and E.Z.; supervision, G.C.; project administration, G.C.; funding acquisition, G.C. All authors have read and agreed to the published version of the manuscript.

**Funding:** This research was partly funded by the Micro Research fund of Wuhan Municipal Bureau of Natural Resources and Planning in 2022, grant number 4.

**Data Availability Statement:** The authors have no data to share.

**Acknowledgments:** The authors would like to thank the editors and reviewers for their support with this manuscript.

**Conflicts of Interest:** The authors declare no known conflicts of interest.

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
