# Peer review of "Spatiotemporal Variation and Influencing Factors of Grain Yield in Major Grain-Producing Counties: A Comparative Study of Two Provinces from China"

_land, doi:10.3390/land12091810_

Round 1

Reviewer 1 Report

This manuscript studied the spatial-temporal evolution and main influencing factors of grain production in Henan Province and Heilongjiang Province from 2000 to 2021 using various methods, and compared the similarities and differences of grain production in the two provinces. However, improvements are still needed in the following aspects:

(1)     Keywords should be the same in the manuscript, such as grain production or grain yield, arable land or farmland. Fix the spelling of “summary” in Abstract.

(2)     Adjust the linguistic logic of the first paragraph of Introduction. Revise the “contradictions” in line 40 to contradictions between agricultural development and ecological environment protection. The first half of the last sentence should be moved to after the third sentence.

(3)     Describe the problems faced and the measures taken by China in the second paragraph of Introduction. Since the first paragraph is the importance of food security in China, the second paragraph should continue with the measures of China to ensure logical continuity.

(4)     Emphasize the superiority of county-scale and comparative analyses in the third paragraph of Introduction. Depict the lack of county-scale studies and comparative analysis studies of grain production in previous studies, and add to their importance respectively.

(5)     Solve the problems of the figures and titles. Supply the altitude units in the legend in Figure 1. The title of Figure 12 does not correspond to the picture. The title of 2.5.1 is wrong.

(6)     Add a paragraph with a description of all the data types used in the Research Data.

(7)     Add specific process for climate data to ensure that slope data and climate data are presented in the same way.

(8)     Parts 3, 4 have a lot of redundancy. The results of the study should be distinguished from the methods, and the methods should not be repeated in the results. The figure of Coefficient of Variation figure has the same trend as the figure of proportion of grain production, so it is sufficient to retain only the former.

(9)     Lack of comparative analysis in Discussion 5.1. This section only describes the situation in the two provinces separately and lacks a comparative analysis of the spatial and temporal evolution of grain production in the two provinces.

(10)  Revise Discussion 5.2. This section should only depict the mechanism of influence of the main influencing factors found in the results.

The problems of English expression in this manuscript mainly include spelling errors, terminology inconsistency, and too many long and complex sentences. Specific modifications have been written in the Comments and Suggestions for Authors, and it is hoped that the authors will carefully revise the sentences in the manuscript.

Author Response

Thanks!

Reviewer 2 Report

The manuscript entitled : "Analysis of spatial-temporal variation from 2000 to 2021 and influencing factors of grain production in major-grain producing counties: A compared study of Henan Province and Heilongjiang Province, China" is evaluating factors that influence grain production in major areas that produce grain crops and this study is crucial for sustainable food production and food security. The research study will be beneficial for farmers, extension officers, policy makers, researchers, students and other readers. I suggest the following to improve the quality of this manuscript:

1. The Abstract lacks the results in the form of the summary of the main findings of the study. To address this, authors need to rephrase the sentence which start from Line 19 to Line 20. Instead of referring to " the following conclusions" the author must talk about the "main findings of the study".

2. Authors need to look at the sentence starting from Line 44 to Line 48 and correct it with specific part under Line 47.

3. Subsection of "Research Areas" under section "Materials and Methods" must be changed to "Description of the Study Areas".

4. Authors need to include general soils that dominate in the two study areas, as land quality and its productivity is also influenced by type of soil dominating in that land.

5. There is no "Results" Section in this manuscript. To address this, Authors need to change section "Analysis of Grain Production Evolution Characteristics" (under Line 279) to "Results" and then Authors continue with the Subsections.

6. Under subsection 3.1.2 authors need to remove sentence under Line 394 starting with "The following conclusions can be drawn". This is because these are not conclusions but are still part of the "Results"

7. Section 4 which is "Analysis of Factors Influencing Grain Production" must be removed as this is still part of the "Results".

8. Authors need to move sentences under Line 1031 to Line 1036 and move these to Materials and Methods. In addition, authors need to remove numbers that appear against paragraphs under Conclusions.

The quality of English Language is relatively fine but minor editing will be required.

Author Response

Thanks!

Reviewer 3 Report

The paper presents a very interesting and solid study but it is a very difficult text to read as it is full of redundancies. The climatic characteristics of two provinces is repeated several times, the there is not much difference between the discussion and conclusion as well as recommendations. There are no references to other studies in the discussion. There is a need to carefully rewrite the text to make it more readible - example: Sentence starting in line 894: there is no need to repeat "regression coefficients for effective irrigated area". Lines 916-917: There is no need to repeat in this sentences "of grain production". The reader knows from the very beginning that the paper is about the grain production 289 times!

> 1. What is the main question addressed by the research?

As stated in the title: "Analysis of Spatial-Temporal Variation from 2000 to 2021 and Influencing Factors of Grain Production in Major Grain-Producing Counties: A Compared Study of Henan Province and Heilongjiang Province, China"

> 2. Do you consider the topic original or relevant in the field? Does it address a specific gap in the field?

I do not consider the topic particularly original but it is highly relevant and interesting methods are used. There are no specific gaps addressed. The study simply invastigates the factors influecing grain production which needs to be regularly analised.

> 3. What does it add to the subject area compared with other published  material?

Current knowledge.

> 4. What specific improvements should the authors consider regarding the methodology? What further controls should be considered?

As mentioned in my review, shorten the text!!!!!!!!!!!!!!!!!!!!!!!!!!!!!!!!!!!!!!!!!!!!!!!!!!!!!!!!

> 5. Are the conclusions consistent with the evidence and arguments presented and do they address the main question posed?

I have already expressed my opinion about it in the review!!! Yes, they are.

> 6. Are the references appropriate?

Yes, they are.

> 7. Please include any additional comments on the tables and figures.

The figure and tables are correct. 

Author Response

Thanks!

Round 2

Reviewer 2 Report

The authors have attended to and have address all my comments.

The quality of English is fine but minor editing are required.